# Electrospun Fibers and Sorbents as a Possible Basis for Effective Composite Wound Dressings

**DOI:** 10.3390/mi11040441

**Published:** 2020-04-22

**Authors:** Alan Saúl Álvarez-Suárez, Syed G. Dastager, Nina Bogdanchikova, Daniel Grande, Alexey Pestryakov, Juan Carlos García-Ramos, Graciela Lizeth Pérez-González, Karla Juárez-Moreno, Yanis Toledano-Magaña, Elena Smolentseva, Juan Antonio Paz-González, Tatiana Popova, Lyubov Rachkovskaya, Vadim Nimaev, Anastasia Kotlyarova, Maksim Korolev, Andrey Letyagin, Luis Jesús Villarreal-Gómez

**Affiliations:** 1Facultad de Ciencias de la Ingeniería y Tecnología, Universidad Autónoma de Baja California, Valle de las Palmas, Mexico. Blvd. Universitario #1000, Unidad Valle de las Palmas, 22260 Tijuana, Baja California, Mexico; aalvarez33@uabc.edu.mx (A.S.Á.-S.); perez.graciela@uabc.edu.mx (G.L.P.-G.); pazj@uabc.edu.mx (J.A.P.-G.); 2National Collection of Industrial Microorganisms (NCIM), CSIR-National Chemical Laboratory, Pune-411008, Maharashtra, India; syedmicro@gmail.com; 3Universidad Nacional Autónoma de México, Centro de Nanociencias y Nanotecnología, Km. 107, Carretera Tijuana a Ensenada, C.P. 22860 Ensenada, Baja California, Mexico; nina@cnyn.unam.mx (N.B.); kjuarez@cnyn.unam.mx (K.J.-M.); elena@cnyn.unam.mx (E.S.); 4“Complex Polymer Systems” Laboratory, Institut de Chimie et des Matériaux Paris-Est, Université Paris-Est Créteil, UMR 7182 CNRS, 2, rue Henri Dunant, F-94320 Thiais, France; grande@icmpe.cnrs.fr; 5Department of Technology of Organic Substances and Polymer Materials, Tomsk Polytechnic University, 634050 Tomsk, Russia; pestryakov2005@yandex.ru; 6Escuela de Ciencias de la Salud, Universidad Autónoma de Baja California- Campus Valle Dorado, Carretera Transpeninsular S/N, Valle Dorado, 22890 Ensenada, Baja California, Mexico; juan.carlos.garcia.ramos@uabc.edu.mx (J.C.G.-R.); yanis.toledano@uabc.edu.mx (Y.T.-M.); 7Facultad de Ciencias Químicas e Ingeniería, Universidad Autónoma de Baja California, 21500 Tijuana, Baja California, Mexico; 8Research Institute of Clinical and Experimental Lymphology – Branch of the Institute of Cytology and Genetics, Siberian Branch of Russian Academy of Sciences, 630060 Novosibirsk, Russia; argentum.popova@mail.ru (T.P.); noolit@niikel.ru (L.R.); nimaev@gmail.com (V.N.); kotlyarova.anastasiya@yandex.ru (A.K.); kormax@bk.ru (M.K.); letyagin-andrey@yandex.ru (A.L.)

**Keywords:** electrospinning, poly (ε-caprolactone), poly (vinyl pyrrolidone), silver sorbents, wound dressings

## Abstract

Skin burns and ulcers are considered hard-to-heal wounds due to their high infection risk. For this reason, designing new options for wound dressings is a growing need. The objective of this work is to investigate the properties of poly (ε-caprolactone)/poly (vinyl-pyrrolidone) (PCL/PVP) microfibers produced via electrospinning along with sorbents loaded with Argovit™ silver nanoparticles (Ag-Si/Al_2_O_3_) as constituent components for composite wound dressings. The physicochemical properties of the fibers and sorbents were characterized using scanning electron microscopy (SEM), differential scanning calorimetry (DSC), Fourier transform infrared spectroscopy (FTIR) and inductively coupled plasma optical emission spectroscopy (ICP-OES). The mechanical properties of the fibers were also evaluated. The results of this work showed that the tested fibrous scaffolds have melting temperatures suitable for wound dressings design (58–60 °C). In addition, they demonstrated to be stable even after seven days in physiological solution, showing no macroscopic damage due to PVP release at the microscopic scale. Pelletized sorbents with the higher particle size demonstrated to have the best water uptake capabilities. Both, fibers and sorbents showed antimicrobial activity against Gram-negative bacteria *Pseudomona aeruginosa* and *Escherichia coli*, Gram-positive *Staphylococcus aureus* and the fungus *Candida albicans*. The best physicochemical properties were obtained with a scaffold produced with a PCL/PVP ratio of 85:15, this polymeric scaffold demonstrated the most antimicrobial activity without affecting the cell viability of human fibroblast. Pelletized Ag/Si-Al_2_O_3_-3 sorbent possessed the best water uptake capability and the higher antimicrobial activity, over time between all the sorbents tested. The combination of PCL/PVP 85:15 microfibers with the chosen Ag/Si-Al_2_O_3_-3 sorbent will be used in the following work for creation of wound dressings possessing exudate retention, biocompatibility and antimicrobial activity.

## 1. Introduction

Appropriate wound management is critical for a suitable healing process; rapid wound closure is desired and is usually achieved using wound dressings. No wound dressings that is ideal for all wound types exists but all of them must fulfill a series of minimal requirements such as: rapid healing, prevention of infection and affordable cost for the patient [1]. Despite the vast offer of wound dressings that exists in the market, highly exuding wounds are difficult to treat and many commercially available medicated dressings cannot effectively prevent the microbial invasion of the wounds [2]. The increased exudate production of some wounds can delay the healing process by slowing or preventing cell proliferation and interfering with growth factor availability. Exudate can also contain elevated levels of inflammatory mediators which are involved in perpetuating wounds, damaging the wound bed, degrading the extracellular matrix and causing skin problems along the wound periphery [3].

The amount of wound exudate produced is partially dependent on the wound surface; therefore, a larger wound surface is expected to produce a higher volume of exudate. Some wound types are perceived to have more copious amounts of exudate [4]. Hence, different kinds of wounds under diverse conditions produce different types and amounts of exudate. It has been reported that burn wounds produce approximately 5000 g/m^2^ of exudate per day, while venous leg ulcers exudate production ranges between 4000 and 12,000 g/m^2^/day [5]. 

The ideal wound dressing should maintain a moist environment at the wound surface while removing excessive exudate. It should allow oxygen/carbon monoxide exchange and act as a physical barrier that prevents infection, avoid adherence to the wound surface, should be made from a readily available biomaterial with minimal processing and must be easily removed without damaging the tissue. Furthermore, the dressing has to be non-toxic and non-allergenic, promote wound healing and contribute with pathogen bacteria elimination [6].

Dressing choice is vital in managing exudate levels, the choice should provide an appropriate moisture balance, avoid maceration of the skin edges, prevent leakage and be easy to apply and remove [7]. It can be mentioned that active prevention of infection can be achieved, among other techniques, by using silver-containing materials [8]. 

Exuding wounds, like burns and ulcers, are particularly complicated to treat and tend to get infected easily. Current treatment protocols usually apply absorbent dressings that are frequently renewed, which injures the underlying newly formed tissue that adheres to the dressings and causes severe pain to the patient when the dressings are removed. This shows the urgent need for wound dressings with the capability to retain wound exudate, prevent infections and that does not harm the new tissue.

Several biopolymers have been proposed for the fabrication of a wound dressing that achieves the aforementioned characteristics, using natural, synthetic, or blend too of them. The use of a polymer blend combines the strength and durability of a specific synthetic polymer with the biocompatibility and bioactivity of certain natural polymers [8,9].

One of the most studied polymers for dressings production is poly (Ɛ-caprolactone) (PCL). PCL is a non-toxic, biocompatible, biodegradable and bioresorbable polymer approved by the FDA for sutures [10], it has also been used as a drug delivery system, as adhesion barriers and wound dressings. Thanks to its interesting mechanical properties [11,12,13], PCL fibers have been combined with collagen, gelatin, hyaluronic acid and ZnO to prepare wound dressings with improved properties. The incorporation of silver nanoparticles provides great results due to the impressive antibacterial properties of silver in its different forms [14,15,16]. Silver ions or Ag nanoparticles have been used in several strategies for wound treatment, among these approaches are included the impregnation of silver particles in polymeric bandages, addition of silver ions in hydrocolloids, foams, alginate dressings and silver sulfadiazine (SSD) creams. All of these tactics promote the antibacterial properties of each application in a wound dealing action [15].

On the other hand, there are a number of approaches that improve the antibacterial properties of the fibrous wound dressings; for example, the addition of chitosan to poly (lactic-co-*glycolic acid*) (PLGA) electrospun wound dressings [17].

Despite the antimicrobial effectiveness observed in different silver formulations *in vitro*, not all the presentations can be used in vivo due to the risk of local argyria, irritation, genotoxicity and hematological effects [18,19]. Nevertheless, a research group has developed a silver nanoparticle formulation known as Argovit™ with a wide antimicrobial spectrum with neither cytotoxic nor genotoxic effects in human lymphocytes and that helps in the rapid healing of diabetic wounds [20,21,22]. International authorities approved this formulation as food additives and for cosmetic and medical devices for veterinary medicine, and human use [22,23,24,25,26,27]. 

Recently, electrospinning has gained special attention for producing micro- and nanofibers with specific properties such as: biocompatibility, stability, versatile morphology, controllable fiber diameter, bioactive surface and desirable degradability depending on the materials used [28,29,30,31,32,33,34,35]. The ultra-fine fibers fabricated by electrospinning are collected in a random pattern, creating fibrous mats that can be used as filter media, catalytic supports, protective clothing materials, cosmetics, sensors, nanocomposites (dental applications), controlled drug delivery systems, medical implants, wound dressings, biosensors and in tissue engineering [32]. Since no reports have been performed yet to combine Argovit™ nanoparticles in electrospun wound dressings, this study explores the potential use of Argovit™ in composite dressing for the treatment of exuding wounds.

This study aims to investigate the properties of electrospun poly (ε-caprolactone)/poly (vinyl pyrrolidone) (PCL/PVP) fibers as a casing for Ag-Si/Al_2_O_3_ sorbents. The combination of an antimicrobial sorbent such as Ag-Si/Al_2_O_3_ and a PCL/PVP scaffold is proposed for the design of an ideal wound dressing. For the readers convenience, the sorbent samples were denoted as Ag-Si/Al_2_O_3_-X, where X is the sorbent number from 1 to 4.

## 2. Materials and Methods

### 2.1. Materials

Poly (ε-caprolactone (PCL) (average MW 80,000 g/mol, Sigma-Aldrich, St. Louis, MO, United States) and poly (vinyl pyrrolidone) (PVP) (average MW 40,000 g/mol, Sigma-Aldrich) were used as provided by distributors, without prior purification for fiber production by electrospinning. Chloroform (ACS grade, Fermont, QC, Canada) was used as a solvent. Purified water was used as obtained from a Millipore RiOs-DI system. The four different Ag-Si/Al_2_O_3_ sorbents were kindly supplied and characterized by Dr. Lyubov Rachkovskaya from Research Institute of Clinical and Experimental Lymphology - Branch of the Institute of Cytology and Genetics, SB RAS, Novosibirsk, Russia. Sorbents were used as received as powdered samples.

Phosphate buffered saline tablets (PBS) biotechnology grade were acquired from Amresco. Human fibroblast HFF-1 cells (American Type Culture Collection- Special Collections Research Center (ATCC)-SCRC-1401), bacterial strains *Staphylococcus aureus* (ATCC 23235), *Escherichia coli* (ATCC 25922), *Pseudomona aeruginosa* (ATCC 15442) and fungal strain *Candida albicans* (ATCC 90028) were purchased from the American Type Culture Collection (ATCC). TOX1 in vitro toxicology assay kit, penicillin-streptomycin, L-glutamine and Dulbecco’s Modified Eagle’s Medium (DMEM) were purchased from Sigma-Aldrich. Fetal bovine serum was acquired from BenchMark, Gemini Bio-Products and sodium bicarbonate and dimethyl sulfoxide (DMSO) from Fermont. Muller-Hilton medium was purchased from Sigma-Aldrich, Difco Sabouraud Dextrose Agar from BD Diagnostic Systems (Franklin Lakes, NJ, USA), and gentamicin and fluconazole from Sigma-Aldrich.

### 2.2. Methods

#### 2.2.1. Sorbent Preparation

Ag sorbent was prepared from silver nanoparticles (AgNPs) solution, named Argovit™, which was kindly donated by Dr. Vasily Burmistrov from the Scientific and Production Center Vector-Vita (Novosibirsk, Russia). Argovit™ represents highly dispersed AgNPs with a total concentration of 200 mg/mL of PVP-covered AgNPs in water. The contents of metallic Ag in Argovit™ is 12 mg/mL, PVP-188 mg/mL.

#### 2.2.2. Electrospinning

Electrospinning is a commonly used technique for producing aligned or non-woven fibrous mats. The final concentration of polymer used to generate all fibers was 13% w/v. A solution of PCL or PVP in chloroform was prepared under continuous stirring (550 rpm) at room temperature for 24 h. The required amount of each polymer to prepare the two polymeric mixtures with composition ratio PCL/PVP w/w 95:5 and 85:15, was dissolved in chloroform followed by stirring (550 rpm) at room temperature for 24 h. Then, 1 mL of the solution was transferred into a 5-mL plastic syringe coupled with a blunt needle (0.8 mm ID). The process was performed as described by Velasco et al. [18]. The loaded syringe was placed in a syringe pump with a programmed flow rate of 1 mL/h, and the process was carried out applying a voltage of 20 kV, with 20 cm distance between the needle tip and the collector, all kept at room temperature at 16–23% relative humidity.

#### 2.2.3. Fibers Characterization

##### Scanning Electron Microscopy (SEM)

The scanning electron microscopy studies were performed in a field-emission microscope JEOL JSM 7600F. Since the samples were not conductive, they required a prior preparation with a thin gold layer. In a metal sample holder, a bit of graphite tape was placed. A small portion of the sample to be analyzed was cut out and placed on the adhesive tape. Then, these samples were coated with a thin layer of metallic gold by plasma-assisted cathodic pulverization, at a voltage around 1 kV for 2 min. After this, the samples were placed in a tray, and electron microscope images were acquired from different areas at different amplification magnitudes. Porosity percentage and average fibers diameter were evaluated by image analysis software ImageJ.

##### Fourier Transform Infrared Spectroscopy (FTIR)

Chemical characterization of samples was accomplished through Fourier Transform Infrared spectroscopy (FTIR) with an ATR-Thermo Scientific Nicolet 6700 equipment. The spectra were collected in the range 400–4000 cm^−1^. The measurements were performed without previous sample preparation.

##### Differential Scattering Calorimetry (DSC)

The TA Instruments DSC Q100 device was used for the evaluation of samples. The study was performed using an aluminum plate for differential scanning calorimetry (DSC), with a temperature range 25 °C to 150 °C and a heating speed of 10 °C/min under a nitrogen atmosphere. The thermogram was analyzed with the software TA Universal Analysis. The weight of the samples ranged from 5 to 10 mg. The study was used to determine the thermal stability and melting temperature (Tm) of the samples.

##### PVP Dissolution Test

PCL, PVP and PCL/PVP fibrous mats produced by electrospinning were used to test the dissolution of PVP in the PCL fibers. Dissolution of PVP electrospun fibers mats was conducted according to Li X et al. [36] and Celebioglu, A. and Uyar, T. [37]. Then, 10 mg of each sample was placed in a glass Petri dish lined with lint-free absorbent paper saturated with 0.9% physiological solution (PISA) at room temperature and were left for seven days. The dissolution of PVP on PCL fibers was recorded with a video camera (Canon PC1304 semi-professional) mounted in a special device that retained light to avoid shadows in the pictures. Experiments were performed by triplicate.

##### Macro-Tensile Measurement on PCL/PVP Fiber Scaffolds

For mechanical testing, the recommendation of the ASTM D882-10 [38] was considered. The fibrous mats were cut with scissors into rectangular samples 0.03 m length × 0.005 m width, a Vernier device was used to measure the thickness after the electrospinning of 3 mL of the polymer solution. Conventional macro-tensile measurements were performed using an electromechanical tensile tester (FG-3000 Digital Force Gauge, NidecShimpo, Glendale Heights, IL, USA). All samples were mounted between holders (Lab made) at a distance of 1 cm. Tensile testing was conducted at a rate of 8 mm/min at room temperature (21 °C). Ultimate tensile strength (MPa), elongation at break (%), experiments were made using five replicates. In the case of the Young´s modules (MPa), yield strength (MPa) a chosen replicate of each sample was used. All data were calculated using the software EDMS-FG V4.6.2 (NidecShimpo). Experiments were made using five replicates.

##### In Vitro HFF-1 Cytotoxicity Test

HFF-1 cells were cultured in Dulbecco’s Modified Eagle’s Medium (DMEM) supplemented with 15% Fetal Bovine Serum (FBS, BenchMark, Gemini Bio-Products), 1% Penicillin streptomycin (Sigma-Aldrich), 1% L-glutamine and 1.5 g/L sodium bicarbonate under incubation conditions of 5% CO_2_ atmosphere, 37 °C and 95% humidity. Subcultures were done each time that culture confluence reached the 80%. 

For cell viability determination, disks of 0.3 cm^2^ from each sample material were cut and sterilized by UV radiation for 15 min. Then, disks were soaked in 100 µL of fresh DMEM media for 1 h and placed into a 96-well plate. After this, 1 × 10^5^ HFF-1 cells per well were seeded on the top of each disc to assure a direct contact with the fibrous scaffolds. After 24 h of incubation, cell viability was determined with TOX1 in vitro toxicology assay kit (Sigma-Aldrich) by a colorimetric assay based on the reduction of methyl-1,3,4-thiazolyltetrazolium (MTT reagent). MTT reagent was added to the plate following the instructions of the manufacturer. The negative control of cell viability was dimethyl sulfoxide (DMSO), which induces total cell death. Cell survival positive control was achieved by incubating the cells only with DMEM media. Experiments were conducted independently by triplicate. Absorbance measurement of MTT reduction was achieved with a 96-well plate reader (Thermo Scientific) at 570 and 690 nm. Absorbance results from the negative control (cell media) were used as 100% of cell survival, survival percentage of each treatment were calculated with %growth = [A_T_/A_C_] × 100, where A_T_ is the absorbance of treated cells, and A_C_ is the absorbance of control cells. Experiments were performed independently in a threefold manner with internal triplicates.

#### 2.2.4. Sorbents Characterization

##### Optical Emission Spectroscopy (ICP-OES)

Sample elemental analysis was performed using an inductive coupled plasma optical emission spectrometer (ICP-OES) Varian Vista-MPX simultaneous CCD. Typically, 125 mg of each sample was digested in the mixture of 2 mL of 63% HNO_3_ and 1 mL of 40% HF, with consecutive dilution in boric acid with total volume 43 mL. All samples were dried at 120 °C for three hours before the acid digestion.

##### Water Uptake Test

In order to determine the water absorption capability of the pelletized or powdered sorbents, a series of gravimetric tests were carried out. The pelletized samples were fabricated by compressing 100 mg of each powdered sample using a commercial pellet press. Phosphate buffer solution (PBS) at 37 °C was used as a wetting agent to emulate the physiological conditions. The tests were carried out following previously reported methods for determining water absorption in pelletized and powdered samples [39,40,41] adapted to the specific requirements of this study. The fluid uptake (%Abs) of the samples was calculated as a ratio of weight increase (W_wet_−W_dry_) to the initial value W_dry_, as described in Equation (1).
(1)$$\%Abs=\frac{\left(W_{wet}-W_{dry}\right)}{W_{dry}}\;\left(100\right)$$

Pelletized pre-weighed dry samples were placed into tissue culture well plates containing 1 mL of PBS at 37 °C. The plates were sealed and placed in an incubator at 37 °C for 2 h. After this time, the samples were taken out of the incubator, removed from the wells and placed onto lint-free absorbent paper towels for 3 s to remove excess liquid; the samples were then weighted. Fluid absorption was determined using Equation (1).

Then, 1 g of each dry sample was placed in a purpose-designed centrifuge basket to measure the fluid absorption of powdered samples, and the total dry weight was registered. The basket was then submerged in a 50-mL beaker containing 10 mL of PBS at 37 °C. The beakers were sealed and placed in an incubator at 37 °C for 2 h. After this time, the baskets containing the samples were removed from the incubator, padded dry on the outside and immediately placed in a centrifuge at 1500 rpm for 30 s to remove excess liquid from the surface of the grains. After this process, the baskets containing the samples were weighted. Fluid absorption was determined using equation 1. ANOVA test (*p* < 0.01) was performed to compare statistically significant differences between water absorption of the samples.

#### 2.2.5. Dressings Assembly

The obtained non-woven fibrous mats produced by electrospinning were cut into 0.032 m length × 0.032 m width squares; the squares were then placed one on top of the other, keeping the edges aligned. The edges were sealed by melting 2 mm of each side using a heating element at 90 °C; before sealing the fourth edge, the pouch was filled with 0.6 g of the sorbent of choice. The amount of sorbent needed was calculated considering the amount of wound exudate production reported by Cutting [5] and two times the water absorption capacity of the sorbent for redundancy.

#### 2.2.6. In Vitro Antimicrobial Test

##### Antibacterial Test

Fiber samples of 0.3 cm^2^ of diameter were cut with a drill, both sides were sterilized with UV-light radiation for 15 min and then placed at the bottom of each well of a 96-well plate. Bacterial strains *Staphylococcus aureus* (ATCC 23235), *Escherichia coli* (ATCC 25922) and *Pseudomona aeruginosa* (ATCC 15442) were cultured in a previously sterilized Muller-Hinton medium at 35 °C for 24 h. After that, bacterial strain population was standardized at 0.5 McFarland scale compared with the standard tube (0.132 abs at 620 nm) with saline solution. Once cell cultures were ready, 150 μL of the clean medium was added to each well-containing the fibers disks of 0.3 cm^2^ or 10 mg/mL of the sorbents. Then, 50 μL of each inoculum with a clean medium was added (*E. coli*, *S. aureus*, *P. aeruginosa*). As a negative control, 150 μL of the clean medium was placed with 50 μL of each strain, without fibers; as a positive control, gentamicin 10 mg/mL was used. All exposed bacterial cells to fibers and sorbents were incubated at 35 °C for 24, 48 and 72 h. As fibers do not dissolve, samples were removed and placed in a new well with the help of sterile forceps, and then washed with 200 μL of the clean medium. Fibers were discarded, and the solution was measured in a Microplate reader (Thermo Scientific) at 620 nm.

##### Antifungal Test

In the case of the antifungal assay, *Candida albicans* (ATCC 90028) were subcultured to DIFCO™ Sabouraud Dextrose Agar (BD Diagnostic Systems, Sparks, NV, USA), at 35 °C (±2 °C). Colonies were picked with a sterile bacteriological loop and suspended in 3 mL of sterile saline solution 0.145 mol/L. The resulting suspension was vortexed for 15 s, and the cell density was adjusted visually to obtain an equivalent transmittance of 0.5 McFarland Standard at a 530 nm wavelength. This procedure provided a standard suspension containing 1 × 10^6^ to 5 × 10^6^ cells/mL. As previously described, 150 μL of the clean medium was added to each well-containing fiber disks of 0.3 cm^2^ or 10 mg/mL of the sorbents. All fungal cells exposed to fibers and sorbents were incubated for 24, 48 and 72 h at 35 °C. Fluconazole at 10 mg/mL was used as a positive control and fungal cell suspension as a negative control. All antimicrobials experiments were done by triplicate.

##### Statistical Analysis

The experiments were done in a threefold independent manner with internal triplicates. The results were expressed as mean ± standard deviation of three independent experiments. Data was evaluated by one way analysis of variance (one-way ANOVA), followed by Tukey’s post-hoc test, using Graph Pad Prism version 6.0c software. The results were considered statistically significant when *p* < 0.05.

## 3. Results

### 3.1. Fibers Characterization

#### 3.1.1. Scanning Electron Microscopy (SEM)

Figure 1 shows the morphology of the electrospun fibers. Generally, images show mostly continuous structure filaments with some irregularities on the PCL fibers surface (Figure 1A) compared with the PVP fibers (Figure 1B). The incorporation of higher amounts of PVP into the PCL structures reduced the surface irregularities observed in pure PCL fibers (Figure 1C,D). For combined PCL/PVP fibers, their diameter increased as higher PVP content was present.

It is important to mention that the showed SEM images were chosen to observe the morphology and rugosity of the fibers (Figure 1). Complete image analysis permitted to obtain diameters distribution of polymeric fibers in histograms, including the average fiber diameter. Each histogram is composed of seven bins or groups that give an idea of the frequency of the fibers’ diameter distribution (Appendix A). Thirty measures in different fields were taken all along the chosen SEM images to make the histograms, with a 200× amplification (Appendix A).

PCL fiber scaffolds presented an average diameter of 2.09 ± 0.78 µm, while PVP fibers possessed diameters of 1.11 ± 0.37 µm, in the case of PCL/PVP fibers; PCL/PVP 95:5 fibers showed a fiber diameter of 39.39 ± 27.63 µm and PCL/PVP 85:15 fibers of 64.94 ± 9.28 µm, respectively.

It is important to note, that the high thickness of PCL/PVP 85:15 fibers (~65 µm) is due to the presence of long bulbs on the fibers, this measurement was taken from the bulbs average diameter, on the contrary, fibers were measured in the parts where no bulbs were present giving diameters of 0.76 ± 0.11 µm (Appendix A).

PCL/PVP fibers demonstrated a higher diameter than their pure polymeric counterparts, and as the amount of PVP was increased, the water absorption of the fibers increased. This can be attributed to the ability of PVP to absorb up to 40% of its weight at ambient conditions [42]. 

On the other hand, PVP-fibers smooth appearance was associated with the small diameter obtained on the fibers. On the other hand, higher amounts of PVP on PCL/PVP fibers resulted in a significant increase in the diameter of the fibers, compared with pure PCL. 

#### 3.1.2. Fourier Transform Infrared Spectroscopy (FTIR)

All samples were analyzed by FTIR spectroscopy to obtain evidence of PVP incorporation into the PCL fibers. Figure 2 shows the spectrum of pure PCL and PVP fibers. Two characteristic absorption bands can be identified for each polymer, C=O vibration frequency of carboxyl group at 1720 cm^−1^ (b) and C-O at 1175 cm^−1^ (c) for PCL and C=O (e) and C-N (d) vibration frequencies of PVP at 1662 and 1423 cm^−1^, respectively. Incorporation of PVP into PCL fibers can be easily monitored by the progressive intensity increase of bands at 1662 (C=O from amide, e) and 1423 cm^−1^ (C-N of tertiary amide, d) as the amount of PVP increases, without intensity decrease of bands corresponding to the vibration frequency of C=O (b) and C-O (c) groups of PCL. Dotted lines in Figure 2 help to identify the absorption frequency of functional groups from PVP on the polymeric mixture in FTIR spectra. 

Finally, other characteristic signals that demonstrate no alteration of the polymers structures corresponding to the C-H alkanes group (a) were observed at 2939 cm^−1^, which form the polymer backbone. 

#### 3.1.3. Differential Scattering Calorimetry (DSC)

A fundamental parameter in a wound dressing scaffold is the thermal stability of the polymeric fibers; it has been reported that burn injuries can reach temperatures up to 41 °C [43,44,45,46]. The reported glass transition temperature of PCL is about −60 °C, while the melting point is situated at 60 °C [43]. In the case of PVP, its melting point is around 150–180 °C [44]. The thermograms of Appendix A show that PCL fibers have a melting point of ~61°C, pure PVP melts at ~148°C, while polymer mixture PCL/PVP with 95:5 and 85:15 ratios have a melting point of ~59°C and ~58°C, respectively. The progressive melting point decrease confirms the presence of PVP in the polymeric mixture as PVP content increases. Melting points for polymers studied in this work are similar with those found in the literature [43,44], identified as appropriate for the application purposes. These thermograms show that the incorporation up to 15% of PVP does not produce a significant modification of the melting point, providing a wide margin of temperature (more than 15 °C) above the burn injuries temperature (41 °C) (Appendix A).

#### 3.1.4. PVP Dissolution Test

As expected, PCL and PCL/PVP fibers showed no evident macroscopic degradation after seven days of observation. On the contrary, PVP fibers were dissolved almost immediately, only 14 s, after being placed on the Petri dish with a saturated physiological solution (Appendix A). SEM images of PCL/PVP fibers taken after seven days of contact of the fibers with physiological solution show the presence of holes all around the PCL fibers, which is a sign of the dissolution of PVP in the polymeric scaffolds. Figure 3 shows SEM images with amplification 10,000× to illustrate PVP dissolution (Appendix A).

#### 3.1.5. Macro-Tensile Measurement on PCL/PVP Fiber Scaffolds

Several mechanical characteristics should be fulfilled for a material to be used as a wound dressing, such as durability, stress resistance, flexibility, pliability and elasticity. In addition, it should be easy to apply and remove without causing any trauma during dressings changes [47]. Therefore, the mechanical properties of the fibrous mats are critical and important to be evaluated.

PCL, PVP and PCL/PVP electrospun fibers were tested using five replicates. Results are reported in Table 1 and Figure 4. It can be observed that all fibrous mats were relatively thin, despite that, mats were manipulable by hand without damage. All fibers presented a thickness of 3 × 10^−4^ m, with the exception of PCL samples which were ~1 × 10^−4^ m thinner after the electrospinning of 3 mL solution. Even when the size of the fibers has an effect on the final thickness of the mat, the adjustment of the mat thickness can be adjusting by increasing the volume (higher than 3 mL) [48].

In the case of the ultimate tensile strength, PCL demonstrated to be more resistant to plastic deformation and ultimate load than PVP [47]. Between two formulations, PCL/PVP 95:5 was more resistant; hence, at a greater content of PCL, better mechanical properties can be found in the samples. As reported by Jeong et al. [49], PCL addition improved the elasticity of PLA films, as can be observed in Figure 5, PCL/PVP 95:5 sample presented a major elastic range and a higher elastic deformation in comparison with PCL/PVP 85:15 sample.

Moreover, elongation at break expresses the capability of the fibrous mat to resist changes of shape without crack formation [50]. PCL/PVP 95:5 fibers presented the higher elongation at break, and PVP fibers demonstrated the lowest one.

All samples presented values of Young’s module that are less than 1 MPa. The PCL/PVP 95:5 mats demonstrated to have the most elasticity (~107 ± 4% elongation at break), and PVP fibers the least elasticity (~21 ± 8% elongation at break) [50].

Finally, Figure 5 shows that PCL/PVP fibers have a higher percentage of a strain compared to PVP and PCL fibers. On the contrary, PVP and PCL needed a higher force to break the fibrous mats.

#### 3.1.6. In Vitro HFF-1 Cytotoxicity Test

One of the great challenges of fibers for biomedical applications is the cytotoxic effect that they can exert on healthy cells. The proliferation of HFF-1 cultures demonstrated that PCL fibers promoted proliferation by 177 ± 22%, and PVP fibers showed a proliferation of 136 ± 35% compared with negative control (DMEM: cell suspension in media). Despite these results, the standard deviations of these samples are high and no significant difference exist when compared to normal growth. Furthermore, the combination of PCL/PVP does not favor the proliferation as considerably as the pure polymeric fibers but not decrease the cell viability compared to the negative control. In the case of PCL/PVP 85:15, the proliferation percentage is 108 ± 23%. Meanwhile, PCL/PVP 95:5 fibers showed lower viability with 69 ± 28% (Figure 4). The observed effects are comparable to those found for a gel prepared from the human amniotic membrane and Aloe vera extract [51,52]. However, synthetic fibers are less expensive and easier to obtain.

### 3.2. Sorbents Characterization

#### Water Uptake Test

The water absorption test was performed to determine which of the Ag-Si/Al_2_O_3_ sorbents (1–4) show the best absorption capability. Been a sorbent a substance which has the property of collecting molecules of another substance by sorption, and are commonly used as materials for blood purification and hemodialysis [53,54,55,56,57,58,59].

Physicochemical properties of the Ag-Si/Al_2_O_3_ sorbents presented on Table 2 were calculated by Rachkovskaya et al. [41]. It can be seen that the proportion of loaded content of Ag present in the sorbent quantified by optical emission spectroscopy (ICP-OES) from this study, where sample content is 0.01% Ag with the exception of sorbent Ag-Si/Al_2_O_3_-2 (0.003% Ag), the content of Ag in samples is related to its antimicrobial activity [14,15,16]. In the case of the particle sizes was assessed by screen size gradation (0.1–1 mm) [41], being an important parameter for water absorption, due to a reported relationship between the increment of particle size decreases mechanical parameters and increases water absorption [60], the sorbents from this work possess high particle size expecting adequate ability for water uptake. Rachkovskaya et al. [41] studied the porous structure of Ag-Si/Al_2_O_3_ samples using the method of low-temperature (77 K) nitrogen vapor sorption (DigiSorb-2600 Micromeritics unit, Atlanta, GA, USA) using generally accepted methods [61]. Before conducting sorption experiments, the samples were trained in a vacuum of 10–4 mmHg at a temperature of up to 100 °C for 5 h. The values of the specific surface area (S_spec_), volume and pore size were calculated from experimental sorption isotherms using the classical Brunauer-Emmett-Teller (BET) method for the range of relative vapor pressures of nitrogen sorbate 0.08–0.25. The preferred pore size in sorbents is 10–100 nm. Samples Ag-Si/Al_2_O_3_-3 and Ag-Si/Al_2_O_3_-4 contain a larger number of thin pores (smaller pores), which provide a higher sorption activity of these samples with respect to water. With respect to the bulk density of the Ag-Si/Al_2_O_3_ sorbents, it has been reported that higher bulk density lowers the ability to absorb water, therefore the sorbent Ag-Si/Al_2_O_3_-3 with less bulk density (0.7 g/cm^3^) is expected to have best water uptake from all samples [62].

As the particle size of the sorbents increased, the total number of pores and the specific surface also increased, regardless of the percentage of silver added. 

The specific surface and the total number of pores of the sorbents play an essential role in water uptake. Figure 6 shows the percentages of water uptake for the pelletized and powdered samples. Regardless of whether the sample is compressed or powdered, the highest water absorption occurred in those samples with the largest specific surface area and with the most significant number of pores. However, when sorbents were pelletized, the water uptake did not decrease significantly. Compressed Ag-Si/Al_2_O_3_-2 sorbent had the lowest water uptake capability (19.8%), while sorbent Ag-Si/Al_2_O_3_-4 absorbed the most amount of water (37.3%) in the compressed pellets study. In the case of the powdered sorbents, as illustrated in Figure 6B, sample 2 demonstrated to have the lowest water absorption capability (30.3%), while sample 4 showed the best absorption rate (43.2%). 

In this study, it can be observed that sorbents Ag-Si/Al_2_O_3_-3 and Ag-Si/Al_2_O_3_-4 in form of pellets or powder possess a better water uptake, that can be attributed to their properties such as: particle size, specific surface and the number of pores (Table 2). Ag amount present in the sorbent does not influence the water uptake. Statistically (ANOVA test, *p* < 0.01), neither pelletized or powdered sorbents showed any significant difference of water absorption.

Compressed pellets are better for a wound dressings application due to the dimensions required in some types of injures. This work shows that pelletizing the samples has no significant effect on water uptake. Therefore, sorbent Ag-Si/Al_2_O_3_-3 and Ag-Si/Al_2_O_3_-4 are the best candidates for the composite dressings fabrication.

### 3.3. Antimicrobials Tests

#### 3.3.1. Antibacterial Test

Gram-negative bacteria *Pseudomona aeruginosa* and *Escherichia coli* and Gram-positive bacteria *Staphylococcus aureus* were chosen to evaluate the antimicrobial activity of the fibers and sorbents. The results showed that both sorbents and fibers produced a decrease in the bacterial population. Fibers presented selective antimicrobial activity depending on their constitution. All samples (fibers and sorbents) showed significant differences in growth at 24 and 48 h, and there was no significant difference at 72 h. On the other hand, the microbial growth of the four-evaluated species presented a statistically different growth after 24, 48 and 72 h of incubation. At 24, 48 and 72 h, the fibrous scaffolds did not cause microbial growth with significant differences (Appendix A). PVP showed no antimicrobial activity on Gram-positive nor Gram-negative strains. Meanwhile, PCL decreased 30%–35% *P. aeruginosa* and *S. aureus* bacterial titer through time and reduced the growth of *E. coli* intermittently, decreasing bacterial population at the first and at the last evaluation time, 24 and 72 h. For fibers from the polymeric mixture, the amount of PVP increased, the higher antimicrobial activity was observed on *P. aeruginosa* and *S. aureus*. PCL/PVP 85:15 also showed antimicrobial activity on *E. coli* intermittently as PCL was the primary proliferation inhibition observed for the evaluated fibers in bacterial cultures. Finally, the bioactivity of the antibiotic was significant with respect to the bioactivity of the fibrous scaffolds at 24, 48 and 72 h (Appendix A).

In the case of sorbents, all of them showed antibacterial activity that increased with exposure time. At 24, 48 and 72 h, the sorbents did not cause growth or decrease in microbial proliferation significantly. Compared to average microbial growth, fibrous scaffolds considerably altered proliferation at 24 h but not at 48 h and 72 h. Moreover, compared to average microbial growth, sorbents did not alter proliferation at 24, 48 and 72 h. Finally, the bioactivity of the antibiotic was meaningful with respect to the bioactivity of the sorbents, at 24 and 48 h, but showed no difference after the 72 h of incubation.

The most significant inhibition growth was demonstrated by sorbents Ag-Si/Al_2_O_3_-1, Ag-Si/Al_2_O_3_-2 and Ag-Si/Al_2_O_3_-3 on the Gram-negative bacteria *P. aeruginosa.* The inhibition growth is 21–28% after 24 h of exposure and reached 58%-66% after 72 h of exposure. Before 72 h of exposure, growth inhibition was observed over time, but without reaching 50% of growth (Figure 7).

*Staphylococcus aureus* and *Escherichia coli* presented more resistance to the exposure of the samples, but still, the presence of the scaffolds and sorbents after 72 h avoided their optimal growth (Appendix A). PCL/PVP 85:15 fibers were more effective against *Staphylococcus aureus*, but PCL/PVP 95:5 fiber presented better bioactivity against *Escherichia coli*. In the case of the sorbents, sorbent Ag-Si/Al_2_O_3_-1 presented acceptable bioactivity against both bacterial strains after 72 h of incubation.

#### 3.3.2. Antifungal Test

For *Candida albicans*, PCL/PVP 85:15 fibers resulted as the most optimal fibers to reduce the growth of this fungal strain. Allowing a cell viability of just ~28% compared to the average growth after 24 h of exposition, also PCL/PVP 95:5 scaffolds were very efficient in diminishing the fungus proliferation affecting more than the half of the population after 24 h. After that time, *Candida* started to adapt to the fibers. Unfortunately, none of the sorbents were effective against the tested *Candida* at the studied silver concentrations (Appendix A).

Despite the low antimicrobial activity displayed by fibers and sorbents compared with the reference compounds, it represents a substantive advantage for dressings fabrication. Fabricated dressings not only will accomplish its task to avoid infections by isolating wounds but also could contribute to eliminating microorganisms in the case of infection in conjunction with systemic antibiotic treatment.

## 4. Discussions

PCL is a material that is extensively used to fabricate scaffolds for tissue engineering due to its biocompatibility, biodegradability, structural stability and mechanical properties. Nevertheless, PCL is known to have low bioactivity and surface energy, which tends to decrease cell adhesion [63]. Although cell affinity is a desired property in a scaffold for tissue engineering applications. The partial hydrophobicity of PCL prevents the adhesion of the newly formed tissue on the surface of the dressings, which helps to prevent further wound damage during removal and favors its regeneration [64]. 

It is well known that PCL has a slow degradation rate [64,65]. Mim et al. [66] tested PCL electrospun nanofibers for degradation after 12 weeks, and non-obvious morphological changes were appreciated. Furthermore, no apparent mechanical properties were perceived once the membranes were dried, although no mechanical studies have been done so far. The slow degradation of PCL fibers is an adequate quality of the dressings proposed in this study. On the other hand, PVP is a known hydrogel capable to absorb a high volume of water [67]. Additionally, PVP is meant to provide the dressings with a soft surface that does not adhere to the wound. 

PCL/PVP scaffolding was firstly studied to improve the low biodegradation rate of pure PCL scaffolds. Kim and co-workers tested different PCL/PVP ratios showing that effectively, upper concentrations of PVP cause faster degradation of PCL/PVP scaffold. As the PVP amount increases, the faster it may be released from the scaffold. PCL/PVP 50:50 ratio scaffold released 97% of the PVP within the first two hours. In addition, they found that average fiber diameters are within the range 1.27–1.88 µm and melting temperatures (T_m_) between 53 and 54 °C [68]. 

Characterization of PCL/PVP fibers scaffolds obtained by electrospinning in this work shows that a higher amount of PVP causes a thickening of the fiber, obtaining diameters 20 to 30 times thicker than the fibers obtained with pure PCL (2.09 ± 0.78 µm) or PVP (1.11 ± 0.37 µm). Average diameters obtained with PCL/PVP 95:5 and 85:15 were 39.39 ± 27.63 µm and 64.94 ± 9.28 µm, respectively. The melting temperatures of PCL and PCL/PVP 95:5 and 85:15 fibers are 61.3, 59 and 58.3 °C. The melting point of fiber studied in the present work is 4–5 °C above those found by Kim.

Differences found between fiber-scaffolds in this work and those obtained by Kim could be associated with the electrospinning conditions. The electrospinning conditions used by Kim to produce PCL/PVP scaffolds with 90:10 and 80:20 w/w ratios, voltage 10 kV, the working distance of 10 cm and flow rate of 0.75 mL/h. Meanwhile, the optimized conditions to produce all scaffolds in this work were a concentration of 13%, voltage 20 kV, flow rate of 1 mL/h and the working distance of 20 cm. Furthermore, it is essential to note that PVP type of molecular weight could also contribute to the differences. Kim used a PVP with a molecular weight of 360 kDa while we used a 40 kDa one. 

An essential factor to consider in the fabrication of polymeric scaffolds is their thermal stability. Dini et al. [69], pointed that the wound bed temperature range was between 31 °C and 35 °C after dressings removal and the perilesional skin temperature was between 31 °C and 34 °C. It is known that the highest temperature reached in burn injuries is 41 °C [40,41]. The melting temperature of all polymeric scaffolds studied in this work was at least 18 °C above this temperature. 

The thermal stability is not the only consideration to produce dressings scaffolds. Low-degradation kinetics of PCL in vivo [70] make an excellent physical barrier between the sorbent and wound surface. In this sense, PCL and PCL/PVP membranes showed no evident macroscopic damage after seven days of submerged in physiological solution. The microscopic analysis showed that PVP release produces holes in the polymeric scaffold. Moreover, the generation of pores increases fibers interconnectivity in the scaffold, which dictates the degree of cellular infiltration and tissue ingrowth, influencing a vast range of cellular processes. In addition, the pores are crucial for diffusion of nutrients, metabolites and waste products, or in this case the wound exudate [68,71,72,73,74,75,76,77]. In addition, PVP release could confer a softer surface of the mats, being a comfortable characteristic for the user, because no further stress is given to the site of the wound. In the present work, a wound dressing was intended to be used less than a week, so the life rate of mats is more than enough for the average application time. 

A prerequisite to use of electrospun nanofibrous scaffolds in biomedical applications is their adequate mechanical properties [78,79]. The strength and deformability of nanofibers have demonstrated to influence in vitro cell migration, proliferation and differentiation, along with cell morphology. The structural integrity and the mechanical strength of the scaffold is essential in the formation of the new tissue [80]. 

In this study, Young´s modulus values of PCL samples were less than (0.32 MPa) compared to literature, where the same fibers presented value 3.8 ± 0.8 MPa. In the case of the average value of strain at break, samples in this work presented a value of 95 ± 16% lower than the reported PCL fibers (170 ± 10%) [78]. These differences are attributed to the thickness of the tested fibrous mats and the molecular weight of the polymer used. In comparison, Young’s modulus of extruded PCL films (molecular weight 80,000 g mol^−1^), determined by macro-tensile testing (Instron), is 190 ± 6 MPa. The differences between Young’s modulus of PCL fiber scaffolds and the extruded films may originate due to the porosity of fiber scaffolds, interactions between fibers (slip of fibers over one another, point bonding, crosslinking) and fiber orientation [81].

In the case of PVP electrospun fibers reported in the literature, the % of elongation at break is 9.10 ± 0.2%, and the ultimate tensile strength is 2.30 ± 0.2 MPa [82]. In case of the electrospun fibers in this work, the % of elongation at the break of PVP samples was 21 ± 8%, higher than those presented in the literature, and the ultimate tensile strength of PCL samples was 2.3 ± 0.2 MPa, which was the same as reported in [83]. PCL/PVP fibers presented better elongation properties than PCL and PVP controls, but less force is needed to break them.

The Young’s modulus measured for the fiber scaffolds remains relevant for their use as wound dressings, as it predicts the mechanical response of the non-woven scaffold in contact with adjacent tissues. Finally, the mechanical properties of single nanofibers within the scaffolds are also crucial in the understanding of the cells behavior when they will be attached to the fiber as they are developing new tissue [78].

Nevertheless, since Young’s modulus of the human skin fluctuates between 0.42 MPa and 0.85 MPa [82], in the present work mats possess the necessary mechanical characteristics to be used as a wound dressing. In the case of films, they are reported to be stronger than fibrous mats. For example, gellan gum films incorporated with Manuka honey (GEL-H) showed up to 38.9 ± 2 MPa of Young’s modulus and they are proposed as wound dressings, but these films showed less elasticity and fragility than fibers in this study [83,84].

PCL and PVP have been reported to be biocompatible polymers and have been used in tissue engineering and drug delivery systems [23,84,85]. The biocompatibility of the fibers is crucial to promote skin wound healing since no cellular stress or infection should be present at the injured site [86]. It has been reported that PCL membranes promote the proliferation of colon cancer cells. By incorporating chitosan and lecithin in the PCL membrane, the proliferation does not decrease, but when gelatin is incorporated, the proliferation is the same as when the membrane is not present [87]. On the other hand, Rogero et al. evaluated the cytotoxicity effects on skin by degradation and release of PVP and low molecular weight compounds. For that, the authors performed a cytotoxicity assay exposing NCTC clone 929 cells to PVP fragments and performed an irritability test by contact in New Zealand rabbits. They concluded that assessed polymeric materials did not present a toxic effect, demonstrating that these biomaterials can be used for the treatments of wounds in the form of advanced bandages or drug immobilization systems as transdermal therapeutic systems [88].

In order to establish the cytocompatibility of polymeric scaffolds with HFF-1 fibroblast cells, the cellular viability was determined after 24 h exposure to the polymeric scaffolds. Figure 4 shows that practically all polymeric scaffolds present no cytotoxicity. Despite the viability decrease observed in Figure 6 for PCL/PVP 95:5, no statistically significant difference was found compared with negative control (cell cultures with DMEM). Other groups reported efficacy of wounds healing using tetrahydro-curcumin even when the viability of HFF-1 cultures decrease by 60%. Therefore, more studies with PCL/PVP 95:5 must be done to completely discard this scaffold [51].

As mentioned before, PCL have been reported as a low bioactive polymer [63]. Despite that, cell growth found for fibroblast exposed to pure PCL scaffolds is in accordance with growth promotion provoked by this polymer in other cells [88]. Some evidence of cell growth promotion is given for pure PCL, for example, it was reported that the proliferation rate of the PCL exposed HDF fibroblasts is higher than the normal growth tested after 3 days [89], moreover Safaeijavan et al. [90] reported an increased proliferation of human fibroblast after been exposed to PCL electrospun scaffolds at 1–6 days.

The results of this work showed that PVP added to PCL reduces growth promotion but not interfere with cell proliferation. The PCL/PVP 85:15 scaffold exemplifies the above (Figure 4). 

One of the most important contributions of this work is the identification of the antimicrobial properties of the PCL/PVP 85:15 scaffold. Two Gram-negative bacteria such as *Pseudomona aeruginosa* and *Escherichia coli*, and one Gram-positive bacteria *Staphylococcus aureus* were chosen to test antibacterial bioactivity of the PCL/PVP fibers and sorbents. These bacterial strains were used because they can cause skin infection with severe damage, such as the case of *Pseudomona aeruginosa* [88,91], *Escherichia coli* can be isolated from surgical and traumatic wounds, foot ulcers and decubitus [92] and *Staphylococcus aureus* are related with skin and soft tissue infections in ambulatory care [93]. For antifungal bioactivity, *Candida albicans* was chosen because diverse fungal species commonly colonize the human skin. Some *Candida* species, especially *C. albicans*, do not only reside on the skin surface as commensals but also cause infections by growing into the colonized tissue [94,95].

Despite the fact that PCL and PVP are polymers with no antimicrobial properties, our results showed that pure PCL in the form of fibers show slight antimicrobial activity for *S. aureus* and *E. coli* (Appendix A); similar results were previously reported by Lagarón and his group [96]. Some evidence can be found that PCL and PVP are not ignored by bacterial cells and the growth is altered, after certain time bacteria recover their viability. Examples of this phenomenum can be seen in [97], where the bacterial growth of the PCL’s exposed *E. coli* is lower than control growth of *E. coli* during 25 h of control. In another study, electrospun fibers of pure PCL showed certain antimicrobial activity against *E. coli* (ATCC25922) with 16.66% drop of activity after 24 h of contact and with *S. aureus* (ATCC6538) with 13.33% drop of activity after 24 h of contact [89]. Finally, PCL films also showed certain reduction of bacterial forming units of *E. coli* and *S. aureus* [96].

On the other hand, PCL/PVP 85:15 scaffold showed an important antimicrobial activity for *P. aeruginosa* and *S. aureus*, and moderate activity for *E. coli* and *C. albicans*. The improvement of antimicrobial activity for PCL/PVP 85:15 scaffold compared with pure PCL fibers could be associated with the prolonged release of PVP. As observed in tumor cells, PVP could interfere with SOD and other antioxidant enzymes of bacterial and fungus to produce growth inhibition [96,98]. The faster dissolution of pure PVP fibers could be the reason not to observe the antimicrobial activity. 

In the case of the reduction of microbial growth due to the pure PVP fibers presence is confirmed by Adomavičiūtė et al. [99]. It was reported that PVP fibers prepared with PEE solvent presented certain bioactivity against *Candida albicans* ATCC 10231, *Bacillus subtilis* ATCC 6633 and *Bacillus cereus* ATCC 11778. In other study, pure PVP films shows bioactivity against *E. coli* and *S. aureus*. Reducing about ~61% of *S. aureus* growth and ~50% of *E. coli* growth when PVP is used as coated film on plastic and glass [100].

On the other hand, some studies confirm that PCL fibers without any treatment do not provoke a considerable inhibition of *S. aureus* [95,101]. Adeli-Sardou M. et al. (2018), reported that PCL/GEL fiber did not affect biofilm-producing bacteria *S. aureus, P. mirabilis, MRSA* and *P. aeruginosa* [102]. However, PCL has been used as a base material to fabricate antimicrobial nanofibers, such as the case of PCL 10% loaded with organic modified-montmorillonite and curcumin, which revealed high antimicrobial activity [103]. However, *P. aeruginosa* proliferates rapidly, this bacterium secretes extracellular polymeric substances (EPS) after 6 h been exposed to PCL scaffolds, indicating that the first hours after inoculation are critical for inhibiting biofilm formation [104]. The above-described behavior was observed for bacterial and fungal strains exposed to PCL/PVP 85:5 scaffold. Significant growth inhibition occurs after 24 h of exposure, being most evident this effect for *C. albicans* (Appendix A).

Once the polymer scaffolds were evaluated, it was determined the antimicrobial activity of sorbents. Ag-Si/Al_2_O_3_ possesses antiseptic qualities and biocompatibility for biological tissues in animal experiments [41]. They also showed a bactericidal effect against *Escherichia coli* [96] and excellent bactericidal and fungicidal properties [105,106]. As mentioned in the results, all sorbents evaluated in this work present an increase in antimicrobial potency over time, being the most evident effect for *P. aeruginosa*. No direct relationship between particle size, water uptake capability of silver content with antimicrobial activity was found. 

A direct comparison of the antimicrobial activity between sorbents and the reference agents’ gentamicin or fluconazole presented in Appendix A, generates the idea that the formers have very low activity. However, if we consider the gentamicin and silver mol equivalents employed, a very different result is obtained. 4.18 µmoles of gentamicin was used to reduce 92%–96% of the bacterial population. Meanwhile, with 9.3 × 10^−5^ µmoles of silver, the bacterial titter reduction was 50%–60% [41], which is the antimicrobial agent of the sorbents. It can be seen that despite the very low amount of Ag nanoparticles within the sorbents, considering these ratios, a significant antibacterial potency exists. Hence future work must be focused on the generation of sorbents with a higher amount of Ag to improve their antimicrobial activity without affectation to their water uptake capability and their cytocompatibility with fibroblast.

The powdered samples exhibited better water retention than their compressed counterparts, as shown in Figure 6B. This feature can be attributed to a larger surface area that allows better fluid infiltration. Since sorbents Ag-Si/Al_2_O_3_-3 and Ag-Si/Al_2_O_3_-4 absorbed the higher amount of water compared with the other samples, they can be proposed as the sorbents of choice to be used in the final dressings. 

In summary, due to its stability, cytocompatibility and antimicrobial properties, the best polymer scaffold is PCL/PVP 85:15. Furthermore, the best water uptake properties and antimicrobial activity were obtained for sorbent Ag-Si/Al_2_O_3_-3. Therefore, the best wound dressings could be obtained with the combination of PCL/PVP 85:15 polymer scaffold and sorbent Ag-Si/Al_2_O_3_-3 with particle size 0.4–0.1 µm and 0.01% of silver. The only disadvantage of using powdered sorbent in the final dressings is the final rigidity of the system. The user has to be careful in applying the dressing bed correctly over the wound. We are currently working on the generation of the complete device for performing the stability studies and evaluating the cytocompatibility and antimicrobial activity with it. 

Finally, there is no previous report using the silver nanoparticles formulation Argovit™ for a wound dressings system prepared with a composite electrospun microfiber. However, this AgNP formulation has been effective in the rapid healing of diabetic wounds [107,108,109]. Several authors propose that AgNPs’ mechanism of action based on the release of silver ions. However, the ion release depends on the method of obtention and the stabilizing agent employed, which in turn modifies their antimicrobial potential and toxicity. Due to the effectiveness showed by Argovit™ directly on the wound, it is crucial to evaluate its efficiency as a part of a wound dressings to reduce the time for complete healing. As shown in this work, its incorporation enhances the antimicrobial activity of the sorbent. Therefore, the direct application of Argovit™ in the wound and the application of wound dressings containing the sorbent impregnated with silver nanoparticles could represent a complete strategy in the treatment of diabetic foot ulcers. 

## 5. Conclusions

This work evaluated the physicochemical properties, cytocompatibility and antimicrobial activity of polymer scaffold and sorbents that will be part of the wound treating dressings. The PCL/PVP 85:15 fibrous scaffolds obtained by electrospinning possessed adequate mechanical properties, melting temperature and decomposition rate in physiological solution and may be used for dressings production. This scaffold produces no modification on cellular viability of HFF-1 fibroblast after 24 h of exposure compared with the negative control that demonstrates its cytocompatibility in wound healing. In addition, the PCL/PVP 85:15 scaffold showed antimicrobial activity against Gram-positive, Gram-negative bacteria and fungus cultures. All these features combined with the good water uptake of sorbent Ag-Si/Al_2_O_3_-3 and its antibacterial and antifungal activity let us conclude that the combination of the PCL/PVP 85:15 scaffold with sorbent Ag-Si/Al_2_O_3_-3 made to produce a wound dressing will generate an effective wound healing. Therefore, in this work two components for potential composite dressings for exuding wound (one of which is electrospun microfibers with a three-dimensional structure and environmentally stable) were evaluated. These components are non-toxic, flexible, with the ability to avoid infection thanks to the Argovit™ included in the sorbent and can absorb and retain water.

## Figures and Tables

**Figure 1 micromachines-11-00441-f001:**
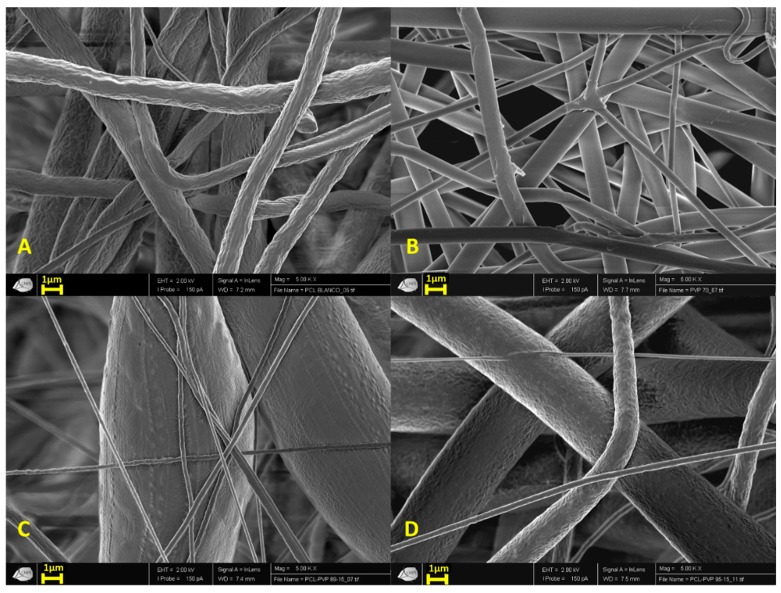
SEM images of (**A**) poly (ε-caprolactone) (PCL), (**B**) poly (vinyl pyrrolidone) (PVP), (**C**) PCL/PVP 85:15 and (**D**) PCL/PVP 95:5 microfibers scaffold. All micrographs are in 5000× amplification (scale bar = 1 µm).

**Figure 2 micromachines-11-00441-f002:**
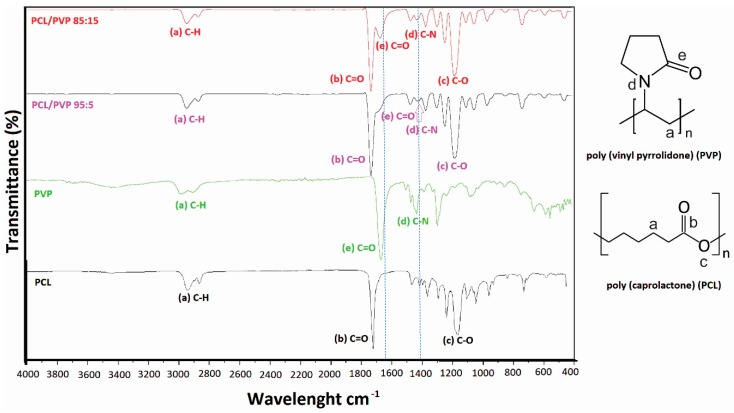
FTIR spectra of PCL, PVP and PCL/PVP fibers.

**Figure 3 micromachines-11-00441-f003:**
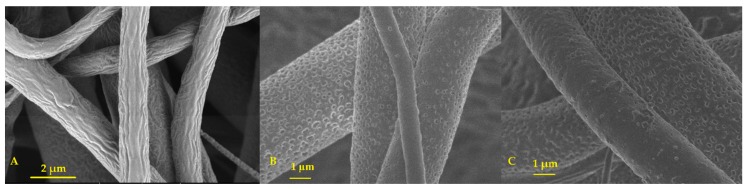
SEM images of (**A**) PCL, (**B**) PCL/PVP 85:15 and (**C**) PCL/PVP 95:5 microfibers scaffold after 7 days of physiological solution contact. All micrographs are 10,000× amplification (scale bar = 2 µm; scale bar = 1 µm for B and C).

**Figure 4 micromachines-11-00441-f004:**
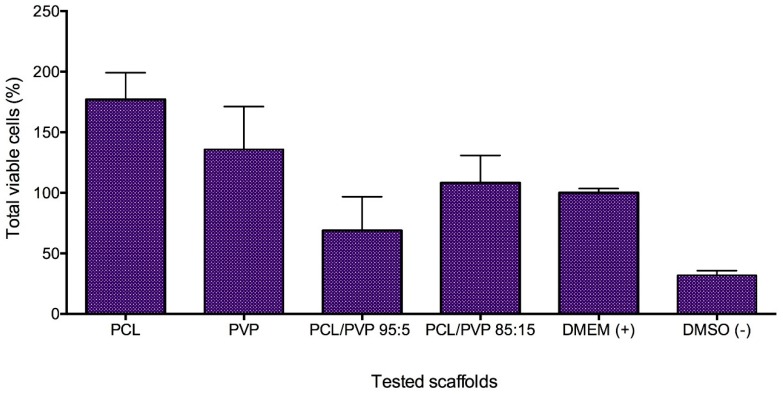
HFF-1 cell viability of electrospun fibers and controls (DMEM as the positive control and DMSO as the negative control).

**Figure 5 micromachines-11-00441-f005:**
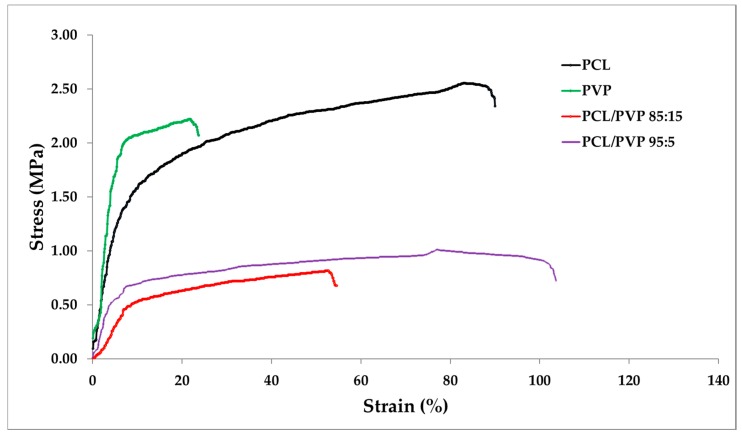
Typical stress–strain curves of electrospun pure PCL, PVP, PCL/PVP composite fiber mats (a selected replicate for each sample was chosen for this graph).

**Figure 6 micromachines-11-00441-f006:**
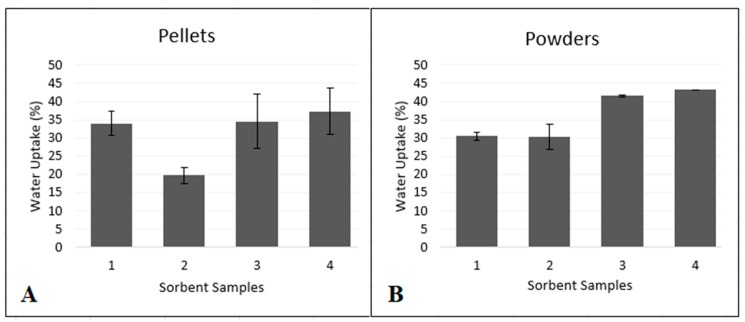
(**A**) Water absorption properties of compressed (pelletized) Ag-Si/Al_2_O_3_ sorbents. (**B**) Water absorption properties of powdered Ag-Si/Al_2_O_3_ sorbents.

**Figure 7 micromachines-11-00441-f007:**
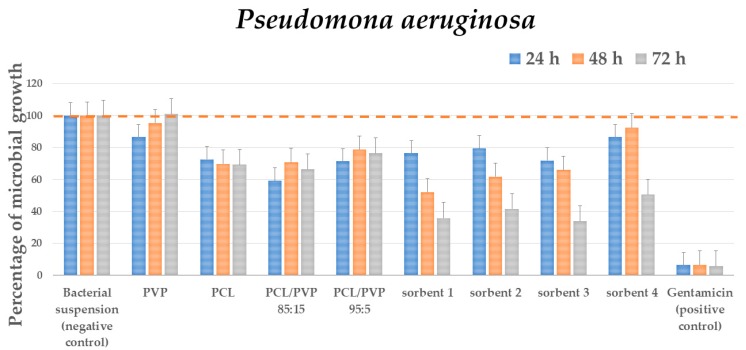
Percentage of microbial growth of exposed *Pseudomona aeruginosa* to PCL/PVP fibers and sorbents, at 24, 48 and 72 h.

**Table 1 micromachines-11-00441-t001:** Mechanical characteristics of fibrous PCL, PVP and PCL/PVP fibers. Thickness (mm), ultimate tensile strength (MPa), elongation at break (%), Young´s modulus (MPa), yield strength (MPa).

Sample.	Thickness (×10^−4^ m)	Ultimate Tensile Strength (MPa)	Elongation at Break (%)	Young´s Modules (MPa)	Yield Strength (MPa)
**PCL**	2.3 ± 1.2	2.5 ± 0.5	94.9 ± 15.7	0.3	1.4
**PVP**	3.1 ± 2.4	2.3 ± 0.2	20.7 ± 7.5	0.4	2.2
**PCL/PVP 95:5**	3.1 ± 1.3	1.2 ± 0.3	107.1 ± 3.6	0.1	0.6
**PCL/PVP 85:15**	3.1 ± 1.2	0.8 ± 0.1	60.6 ± 19.2	0.1	0.5

Data is presented in (average ± standard deviation). Experiments were done with five replicates.

**Table 2 micromachines-11-00441-t002:** Physico-chemical properties of Ag-Si/Al_2_O_3_ sorbents (Based on [41]).

Ag- Si/Al_2_O_3_ Sorbents Number	Ag (%)	Particle Size (mm)	S_spec_ (m^2^/g)	V_∑_pores (cm^3^/g)	P (g/cm^3^)
**Ag- Si/Al_2_O_3_**-1	0.01	0.10	100.0	0.25	0.90
**Ag- Si/Al_2_O_3_**-2	0.003	0.04	96.6	0.20	1.10
**Ag- Si/Al_2_O_3_**-3	0.01	1.00	245.0	0.35	0.70
**Ag- Si/Al_2_O_3_**-4	0.01	0.80	250.0	0.35	0.75

Designations: Spec: Specific surface. V∑pores: Total number of pores. P: bulk density. All sorbents were evaluated as powders.

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
