# Peer review of "Electrospun Fibers and Sorbents as a Possible Basis for Effective Composite Wound Dressings"

_micromachines, 2020, doi:10.3390/mi11040441_

Round 1

Reviewer 1 Report

The paper describes an effort to prepare a composite dressing for burns with exuding burns. The authors propose to use a composite material that is composed of degradable polymers PCL/PVP as the material base and Si/Al2O3 as the sorbent. The composite material is electrospun into mat. The result sample is characterized by SEM, DSC, FTIR, and cell cultures for biocompatibility tests.

The composite material is an interesting concept for the burn dressing application, but the novelty involved in this study is very limited. It is not clear how the DCS and FTIR results are even directly relevant to the burn dressing applications. In stead, some mechanical tests of the resulting mat, and the amount of Si/Al2O3 sorbent incorporated into the composite should be investigated in details to make the paper more relevant and interesting to its readers.   

Author Response

On behalf of all authors, we want to thank the reviewers for their excellent suggestions and comments. Certainly, it will increase and secure the scientific quality of this manuscript.

In this document, we enlisted the comments and suggestions from the reviewers and enclosed a table that specifies the corrections and locations of them.

Point-by-point response

Reviewer 2 Report

  1. There is a lack of innovation in this manuscript. PCL/PVP has long been electrospun for wound dressing application. In fact, due to a long period of degradation process, PCL might not be a good choice. The authors should give a detailed discussion of this choice.
  2. The authors add sorbents to improve its water uptake ability and antibacterial activities. The Ag has been widely used in antibacterial wound dressing. There should be a discussion regarding the sorbents in the introduction section. Different types of nanoparticles have been added in the wound dressing in the literature, for example, the incorporation of Mesoporous silica nanoparticles (MSN) not only improved the hydrophilicity of the wound dressing but also offered the ability as a drug carrier for antibacterial treatment (Jia et al., Nanomedicine, 2018, 13(22): 2881-2899). In fact, there are a number of approaches to improve the fibrous wound dressing antibacterial activities, for example, Chitosan modified PLGA electrospun wound dressing (Yang et al., Polymers, 2017, 9, 697). The authors should give detailed discussions in the manuscript to show the novelty of this work in the context of the aforementioned work.
  3. There is no evidence given for the inclusion of the sorbents in the fibrous dressing, presented in this manuscript, apart from the water uptake test. More tests should be carried out, for example, FTIR and XRD should be performed.
  4. There are some mistakes in the manuscript: the cell viability test was carried out, however, the results of proliferation were given in Figure 6.
  5. The results are not well presented. It would be better to use charts instead of tables for the microbial growth data, which was presented in Table 3. More detailed antibacterial results should be given, for example, the Histogram of bacterial counts of each sample.

Author Response

(The authors gave the same response as above.)

Round 2

Reviewer 1 Report

The authors have addressed my concerns adequately.  

Author Response

Behalf all author Alan Saúl Alvarez Suárez, Syed G. Dastager, Nina Bogdanchikova, Daniel Grande, Alexey Pestryakov, Juan Carlos García-Ramos, Graciela Lizeth, Pérez-González, Karla Oyuki Juarez, Yanis Toledaño-Magaña, Elena Smolentseva, Juan Antonio Paz-González, Tatiana Popova, Lyubov Rachkovskaya, Vadim Nimaev, Anastasia Kotlyarova, Maksim Korolev, Andrey Letyagin, Luis Jesús Villarreal-Gómez, we would like to thank for further revision of manuscript micromachines-732512。

Reviewer 2 Report

I am satisfied with the revision. 

Author Response

Behalf all author Alan Saúl Alvarez Suárez, Syed G. Dastager, Nina Bogdanchikova, Daniel Grande, Alexey Pestryakov, Juan Carlos García-Ramos, Graciela Lizeth, Pérez-González, Karla Oyuki Juarez, Yanis Toledaño-Magaña, Elena Smolentseva, Juan Antonio Paz-González, Tatiana Popova, Lyubov Rachkovskaya, Vadim Nimaev, Anastasia Kotlyarova, Maksim Korolev, Andrey Letyagin, Luis Jesús Villarreal-Gómez, we would like to thank for further revision of manuscript micromachines-732512,